# Design and Synthesis of New Benzo[d]oxazole-Based Derivatives and Their Neuroprotective Effects on β-Amyloid-Induced PC12 Cells

**DOI:** 10.3390/molecules25225391

**Published:** 2020-11-18

**Authors:** Zheng Liu, Ming Bian, Qian-Qian Ma, Zhuo Zhang, Huan-Huan Du, Cheng-Xi Wei

**Affiliations:** 1Medicinal Chemistry and Pharmacology Institute, Inner Mongolia University for the Nationalities, Tongliao 028000, China; Liuzheng0314@163.com (Z.L.); bmz3@163.com (M.B.); maqq2020@126.com (Q.-Q.M.); 2Inner Mongolia Key Laboratory of Mongolian Medicine Pharmacology for Cardio-Cerebral Vascular System, Tongliao 028000, China; 3College of Pharmaceutical Sciences, Yanbian University, Yanji 133022, China; zhangzhuo0523mm@163.com

**Keywords:** synthesis, benzo[d]oxazol, thiadiazoles, Alzheimer’s disease, β-amyloid, Akt/GSK-3β/NF-κB signaling pathway

## Abstract

A series of novel synthetic substituted benzo[d]oxazole-based derivatives (**5a**–**5v**) exerted neuroprotective effects on β-amyloid (Aβ)-induced PC12 cells as a potential approach for the treatment of Alzheimer’s disease (AD). In vitro studies show that most of the synthesized compounds were potent in reducing the neurotoxicity of Aβ_25-35_-induced PC12 cells at 5 μg/mL. We found that compound **5c** was non-neurotoxic at 30 μg/mL and significantly increased the viability of Aβ_25-35_-induced PC12 cells at 1.25, 2.5 and 5 μg/mL. Western blot analysis showed that compound **5c** promoted the phosphorylation of Akt and glycogen synthase kinase (GSK-3β) and decreased the expression of nuclear factor-κB (NF-κB) in Aβ_25-35_-induced PC12 cells. In addition, our findings demonstrated that compound **5c** protected PC12 cells from Aβ_25-35_-induced apoptosis and reduced the hyperphosphorylation of tau protein, and decreased the expression of receptor for AGE (RAGE), β-site amyloid precursor protein (APP)-cleaving enzyme 1 (BACE1), inducible nitric oxide synthase (iNOS) and Bcl-2-associated X protein/B-cell lymphoma 2 (Bax/Bcl-2) via Akt/GSK-3β/NF-κB signaling pathway. In vivo studies suggest that compound **5c** shows less toxicity than donepezil in the heart and nervous system of zebrafish.

## 1. Introduction

Benzo[d]oxazoles are important scaffolds in heterocyclic compounds, which are extensively found in diverse pharmacologically active substances and natural compounds. For instance, benzo[d]oxazole plays an important role as a key building block in β adrenergic receptor antagonists [1], and anti-inflammatory [2], antimicrobial [3], and anticonvulsant [4] agents. Studies in recent decades have indicated that the thiadiazole ring is an important framework with broad-spectrum biological activity [5]. Behjat Pouramiri et al. reported that a series of novel benzo[d]oxazole derivatives have been synthesized as potential inhibitors of acetylcholinesterases (AChE). Some of these compounds were most effective against AChE [6]. For example, donepezil, an FDA-approved drug for the treatment of Alzheimer’s disease (AD), delayed the symptoms of AD by reversibly inhibiting AChE. The thiadiazole ring has been used to link anti-Alzheimer compounds in the past [7,8,9]. Therefore, we selected the compounds **5a**–**5v** for activity analysis of AD.

AD is a neurodegenerative disorder characterized by progressive memory deficits. Some studies predict that by 2030 approximately 65.7 million people worldwide will suffer from AD. If there is still no effective treatment, this number will reach 115.4 million by 2050 [10]. AD is neuropathologically defined as the extracellular accumulation of β-amyloid (Aβ) peptides into amyloid plaques and the formation of intracellular neurofibrillary tangles (NFTs) with hyperphosphorylation of tau protein [11]. Aβ-induced hyperphosphorylation of tau has been recognized as a contributor to AD pathogenesis and progression, and abnormal accumulation of intracellular Aβ aggregates or particles in the brain of AD patients induces cell apoptosis and even directly triggers neuronal cell death [12,13]. Aβ_25-35_ is a toxic peptide fragment of the full-length Aβ peptide, and it could induce a direct toxic effect in nerve cells and lead to their apoptosis [14]. Thus, Aβ plays a crucial role in the pathogenesis of AD.

Aβ could induce neuronal apoptosis by regulating glycogen synthase kinase (GSK-3β) signaling pathways [15]. GSK-3β, an important kinase, is a critical element involved in the regulation of amyloidogenic processing of Aβ. Increased GSK-3β activity has been observed in the brains of patients with AD, and its pathological activation facilitates Aβ production and neuritic plaque formation [16,17] and upregulates nuclear factor-κB (NF-κB) signaling pathways, which eventually leads to cell apoptosis. In the brains of patients with AD, activated NF-κB is observed predominantly in neurons and glial cells in Aβ plaque surrounding areas [18,19]. Inhibition of NF-κB alleviated phosphorylation of tau and neurotoxicity of Aβ in animal models and patients with AD [20]. NF-κB regulates the expression of many molecules in apoptosis, including inducible nitric oxide synthase (iNOS) [21], receptor for AGE (RAGE) [22], B-cell lymphoma 2 (Bcl-2) and Bcl-2-associated X protein (Bax) [23]. iNOS are enzymes catalyzing the production of nitric oxide (NO) from l-arginine, and their expression is frequently associated with inflammation [24]. Moreover, activation of NF-κB also induces Aβ production through the upregulation of β-site amyloid precursor protein (APP)-cleaving enzyme 1 (BACE1) expression [25]. Therefore, the neurotoxicity of Aβ can be inhibited by regulating NF-κB [26]. We have investigated the compound **5c** protected Aβ_25-35_-induced PC12 cell model and examined this protection through Akt, GSK-3β and NF-κB signaling pathways.

## 2. Results and Discussion

### 2.1. Chemistry

Compounds **5a**–**5v** were prepared according to Figure 1. Overall, they were prepared in three steps. The benzo[d]oxazole-2-thiol was treated with bromoacetic acid through the Williamson reaction at 60 °C for 4 h to obtain intermediate **2**. The key intermediates **4a**–**4v** were synthesized from commercially corresponding aniline of phosphorus oxychloride, thiosemicarbazide and substituted benzoic acid. Final product synthesis was performed in parallel using water-soluble EDCI and HOBt to activate the carboxylic acid and treated with the requisite amine in the presence of DMAP. The final product results are shown in Table 1.

The structures of the new compounds were confirmed by spectral data (^1^H-NMR, ^13^C-NMR, and HRMS, see Appendix A).

### 2.2. Effects of Compounds on Cell Viability of Aβ_25-35_-Induced PC12 Cells and Selection of Active Compounds

We first tested the cytotoxicity of 22 compounds. Compounds **5a**–**5c** at the concentration of 30 μg/mL were added to the PC12 cells for 24 h. As shown in Figure 2A, compound **5c**, **5j**, **5l**, **5o**, **5q** and **5t** had no obvious toxic effects on PC12 cells at the concentration of 30 μg/mL (*p* > 0.05).

Then, we tested the protective effects of 22 compounds on Aβ_25-35_-induced PC12 cells. Consequently, 20 μM of Aβ_25-35_ and compounds **5a**–**5c** at the concentration of 5 μg/mL were added together to the PC12 cells for 24 h. As shown in Figure 2B, compounds **5c**, **5o** and **5t** could significantly protect PC12 cells from Aβ_25-35_-induced toxicity (*p* < 0.001).

In order to find the compound with the best protective effect on PC12 cells against Aβ_25-35_-induced toxicity, we tested compounds **5c**, **5o** and **5t** at the concentrations of 0.625, 1.25 and 2.5 μg/mL with 20 μM of Aβ_25-35_ added together to the PC12 cells for 24 h. As shown in Figure 2C, compounds **5c** and **5o** had protective effects on Aβ_25-35_-induced PC12 cells (*p* < 0.001), and their effects were better than compound **5t** (*p* < 0.01) at the concentration of 0.625 μg/mL, as the concentration of 0.625 μg/mL approximately equals the molar concentration 0.001472 μmol/mL in compound **5c**, which is smaller than that in compound **5o**, namely molar concentration 0.001543 μmol/mL. We chose to study compound **5c**’s pharmacological mechanism of protective effect on Aβ_25-35_-induced PC12 cells.

### 2.3. Compound ***5c*** Reduced Tau Phosphorylation, Akt and GSK-3β Activation in Aβ_25-35_-Induced PC12 Cells

In the PC12 cell model induced by Aβ_25-35_, the expression of tau protein was significantly higher than that of the control group, while the expression of Akt and GSK-3β was reduced, which was consistent with previous reports [27]. GSK-3β is one of the primary tau kinases and its activity requires serine dephosphorylation [28], the presence of mutual regulatory systems between kinases including GSK-3β or Akt and phosphatases such as PP2A [29]. Akt regulated the phosphorylation of GSK-3β, thereby participating in tau protein hyperphosphorylation and cell apoptosis [30]. Compound **5c** significantly inhibited the expression of p-GSK-3β/GSK-3β (*p* < 0.001, Figure 3E,G), so did the hyperphosphorylation of tau protein at the Thr181-p-tau, Thr205-p-tau and Ser396-p-tau sites (*p* < 0.01 or *p* < 0.001, Figure 3A–D) and increased the ratio of p-Akt/Akt (*p* < 0.001, Figure 3E,F) together with donepezil. Our results show that donepezil could not inhibit the expression of GSK-3β (Figure 3E,F) alone. The reason for this may be that GSK-3β partly mediated the protective effects exerted by Akt. For example, GSK-3β was regulated by inhibition of the MPTP opening as a consequence of the reduction in the Ca^2+^ overload and by reduced hydrolysis of adenosine triphosphate (ATP) [31,32,33]. The results and mechanism need to be further discussed in the future.

### 2.4. Compound ***5c*** Inhibited Expression of NF-κB in Aβ_25-35_-Induced PC12 Cells

NF-κB is an essential signaling pathway involved in the survival, proliferation, and apoptosis of neurons. The results show that compound **5c** and donepezil significantly reduced Aβ_25-35_-induced p-NF-κB/NF-κB levels compared to control cells (*p* < 0.01 or *p* < 0.001, Figure 4).

### 2.5. Compound ***5c*** Inhibited Expression of Bax and Bcl-2 in Aβ_25-35_-Induced PC12 Cells

NF-κB also regulated the expression of Bax. Bcl-2 belongs to an anti-apoptotic protein family, while Bax belongs to a pro-apoptotic protein family [24]. Compound **5c** and donepezil decreased the expression of Bax and increased the expression of Bcl-2 (*p* < 0.001, Figure 5).

### 2.6. Compound ***5c*** Inhibited Expression of BACE1 in Aβ_25-35_-Induced PC12 Cells

Cleavage of two proteases, β- and γ-secretase, mediated the endo proteolysis of amyloid precursor protein (APP) and finally produced Aβ. BACE1 is considered as the major form of β-secretases. Therefore, reducing the expression of BACE1 has become one of the therapeutic targets for AD [34]. The results indicate that compound **5c** inhibited expression of BACE1 in PC12 cells (*p* < 0.05, *p* < 0.01 or *p* < 0.001, Figure 6). Our results show that donepezil did not inhibit the expression of BACE1 in PC12 cells (Figure 6), and that may cause donepezil to reversibly inhibit AChE and partly inhibit BACE1. The results and mechanism need to be further discussed in the future.

### 2.7. Compound ***5c*** Inhibited Inflammation-Related Factors iNOS of Aβ_25-35_-Induced PC12 Cells

Microglia are considered to be key in innate immune and inflammatory responses in AD. Due to microglial over activation, a wide range of inflammatory cytokines were released, which could lead to the death of neurons [35]. The iNOS were regulated via NF-κB signaling pathway [36]. The Western blotting results demonstrate that compound **5c** and donepezil inhibited the protein expressions of iNOS in Aβ_25-35_-induced PC12 cells (*p* < 0.001, Figure 7). Therefore, compound **5c** may exert neuroprotective effects via inhibiting protein expression of pro-inflammatory cytokines in Aβ_25-35_-induced PC12 cells.

### 2.8. Compound ***5c*** Inhibited Expression of RAGE of Aβ_25-35_-Induced PC12 Cells

RAGE is a member of the immunoglobulin superfamily [37]. The activation of RAGE induces inflammatory responses, and these effects are often mediated by NF-κB-mediated cytokine gene expression [38]. Western blotting results demonstrate that compound **5c** inhibited the expression of RAGE in Aβ_25-35_ induced PC12 cells (*p* < 0.001, Figure 8).

AD is a complex multifactorial pathology. Aβ exhibited neurotoxicity and contributed to neuronal death, which was thought to be the primary factor in initiating the pathogenesis of AD [39,40]. Several neurotoxic effects of Aβ shown in the present study are consistent with previous studies [41,42]. In the present study, Aβ_25-35_ reduced the cell viability of untreated PC12 cells; conversely, compound **5c** increased that of Aβ_25-35_-induced PC12 cells. Tau is the major neuronal microtubule associated protein. In the brain of AD patients, tau is abnormally hyperphosphorylated, accumulating as intraneuronal tangles and failing to maintain structures [43], causing cell apoptosis [44,45,46]. Thus, pharmacologic strategies designed to suppress hyperphosphorylation of tau may be beneficial for the treatment of AD. In the present study, we observed that Aβ_25-35_-induced tau hyperphosphorylation at thr181-p-tau, thr205-p-tau and ser396-p-tau sites was significantly inhibited by compound **5c**.

The Akt/GSK-3β signaling pathway is directly affected by Aβ exposure and plays an important role by apoptosis, and the activity of this pathway is impaired in the AD brain [47]. Under certain conditions, regulation of the Akt/GSK-3β signaling pathway can effectively inhibit neuronal apoptosis. GSK-3β-dependent regulation was imperative to amyloidogenic and hyperphosphorylation of tau [48]. Our findings show that compound **5c** reduced the phosphorylation of Akt and GSK-3β in Aβ_25-35_-treated PC12 cells. GSK-3β activity is abnormally enhanced in AD patients, Aβ inhibition of phosphorylation of Akt and GSK-3β [44] modulated nuclear translocation of NF-κB [49], resulting in increase in the downstream pro-apoptotic factor Bax and decrease in the downstream anti-apoptotic factor Bcl-2, eventually resulting in cell apoptosis and death [50,51,52]. In this study, we observed that compound **5c** reduced the expression of NF-κB and Bax/Bcl-2 in Aβ_25-35_-induced PC12 cells.

Aβ is generated by APP, BACE1 is required for the cleavage of APP to Aβ, and thus represents a prime therapeutic target for AD [53]. Aβ can bind to RAGE and multiple cellular signaling cascades become activated. In addition, RAGE itself is one of the genes that are activated by NF-κB [54]. High expression of RAGE in cells can trigger activation of NF-κB, which in turn up-regulates the expression of RAGE [55,56]. In this study, we observed that compound **5c** reduced expression of BACE1 and RAGE in Aβ_25-35_-induced PC12 cells. The NF-κB transcription factor complex is a pleiotropic activator that participates in the induction of a wide variety of genes and regulation of the generation of inflammatory cytokines, and Aβ deposition promotes the activity of NF-κB in degenerative neurons [57]. In the present study, we observed that compound **5c** prevented Aβ_25-35_-induced expression of NF-κB and iNOS in PC12 cells, indicating that compound **5c** might be beneficial for delaying the progression of inflammatory responses in AD. In summary, compound **5c** had neuroprotective effects on Aβ_25-35_-induced PC12 cells via the Akt/GSK-3β/NF-κB signaling pathway (Figure 9).

### 2.9. Effects of Compound ***5c*** and Donepezil on Zebrafish Mortality and Hatching Rate

No mortality was observed in the 0.2% DMSO treatment. Mortality was observed at 0 to 500 μM for 120 hours post-fertilization (hpf) in compound **5c** and donepezil (Figure 10A,B). At 96 hpf, the mortality of compound **5c** and donepezil of 250 μM reached 20% and 83.3%, respectively, and the mortality in the 500 μM group reached 30% and 100%, respectively. At both time points, donepezil led to a significantly higher mortality than compound **5c**. The LC_50_ of compound **5c** and donepezil at 120 hpf were 1.232mM and 0.135 mM, respectively. The hatching rate was calculated for each exposure group at 96 hpf and the hatching rate of the control group was 100%. For compound **5c**, the hatching rates of 0 and 500 μM were 90% but hatching rates of the 250 and 500 μM in the donepezil groups decreased significantly to 16.7% and 13.3%, respectively (Figure 10C). These results indicate that compound **5c** is less toxic than donepezil.

### 2.10. Effects of Compound ***5c*** and Donepezil on Heart Rate

Compared with the control group, both compound **5c** and donepezil exposure groups recorded a decrease in heart rate (beats per minute). At 96 hpf, for compound **5c**, when the concentration increased from 50 to 100 μM, the heart rate decreased to 96.77% of control to 86.57% of control. Donepezil led to a decrease in heart rate even at 50 μM, where heart rate was 70.14% of control (*p* < 0.01). The heart rate of donepezil exposed individuals decreased in a dose dependent manner and at 100 μM, the heart rate was just 64.18% of the control group (*p* < 0.01, Figure 11F).

### 2.11. Effects of Compound ***5c*** and Donepezil on the Pericardial Cavity Area

Zebrafish pericardial cavity was imaged at 96 hpf, and the area of the pericardial cavity under each treatment was quantitatively analyzed. At 96 hpf, 50 and 100 μM compound **5c** slightly increased the pericardial cavity area. However, 50 and 100 μM donepezil exposure groups recorded a significantly increased area (Figure 11G). Pericardial areas of the donepezil exposed groups were 1.17–1.30 times larger than the corresponding compound **5c** exposure groups at 50 and 100 μM, respectively.

### 2.12. Effects of Compound ***5c*** and Donepezil on Tactile Sensitivity

To assess the effect of compound **5c** and donepezil on the nervous system of zebrafish during early embryonic development, the tactile sensitivity test for 96 hpf was designed. In the tactile sensitivity tests, we found that when treated with 50 and 100 μM of compound **5c** and donepezil, the zebrafish larvae swam out of the bottom circle less than the control group. However, compared to 50 and 100 μM of compound **5c**, the donepezil exposure groups of zebrafish larvae swam out of the bottom circle significantly less times (*p* < 0.001, Figure 12). The results suggest that compound **5c** showed less toxicity than donepezil on the nervous system in the early stage of embryonic development.

## 3. Experimental Section

### 3.1. Chemistry

All reagents and solvents were purchased from commercial sources and used as received without further purification. Reactions were monitored by thin-layer chromatography (TLC) in silica gel and the TLC plates were visualized by exposure to ultraviolet light (254 and 365 nm). Compounds were purified by flash column chromatography over silica gel (200–300 mesh). Melting points (uncorrected) were determined on a RY-1 MP apparatus. Furthermore, ^1^H-NMR spectra were measured on an AV-300 (Bruker, Fällanden, Switzerland), and all chemical shifts were given in parts per million relative to tetramethylsilane.

### 3.2. General Procedure for the Preparation of Compound ***2***

The mixture of benzo[d]oxazole-2-thiol **1** (6.04 g, 0.04 mol), chloroacetic acid (3.78 g, 0.04 mol) and NaOH (3.2 g, 0.08 mol) in 50 mL acetone was heated at 60 °C for 4 h. After cooling to room temperature, the solution was neutralized with hydrochloric acid. The precipitate was filtered and then recrystallized from aqueous ethanol (5:1) to obtain purified **2** as a pale purple solid. Yield 72%.

### 3.3. General Procedure for the Synthesis of Compounds ***4a**–**4v***

To a solution of benzoic acid (244.2 mg, 2.0 mmol), thiosemicarbazide (182.3 mg, 2.0 mmol) and POCl_3_ (1.2 mL) were added and heated at 80 °C for 2.5 h. After cooling to room temperature, water (2.5 mL) was added. The reaction mixture was refluxed for 4 h. After cooling, the mixture was basified to pH = 8 by the dropwise addition of 40% NaOH solution under stirring. The precipitate was filtered and purified through column chromatography to give pure **4a**–**4v** in 74–88% yield.

### 3.4. General Procedure for the Synthesis of Compounds ***5a**–**5v***

To a solution of compound **2** (104.6 mg, 0.5 mmol) in DCM (20 mL), the **4a**–**4v** (1.0 mmol), HOBt (0.26 mmol), EDCI (0.55 mmol), DMAP (0.55 mmol) and triethylamine (0.55 mmol) were added. The mixture was stirred under nitrogen at room temperature for 12 h. When TLC showed the reaction was completed, the reaction mixture was poured into water and extracted with DCM. The residue was obtained by evaporation of the solvent and purified by flash chromatography to give the pure product. Yields of purified compounds ranged from 43% to 64%.

*2-(benzo[d]oxazol-2-ylthio)-N-(5-phenyl-1,3,4-thiadiazol-2-yl)acetamide* (**5a**). Yellow solid; yield: 64%, m. p. = 260.2–261 °C. ^1^H-NMR (300 MHz, DSMO-*d*_6_) δ 13.14 (s, 1H), 7.99–7.87 (m, 2H), 7.69–7.59 (m, 2H), 7.57–7.47 (m, 3H), 7.32 (dd, *J* = 5.5, 3.7 Hz, 2H), 4.53 (s, 2H). ^13^C-NMR (75 MHz, DSMO-*d*_6_) δ 166.07, 163.33, 162.15, 158.33, 151.37, 141.09, 130.66, 130.00, 129.33, 126.92, 124.69, 124.42, 118.31, 110.26, 35.38. HRMS (ESI) *m*/*z* calcd. for C_17_H_12_N_4_O_2_S_2_Na^+^ [M + Na]^+^ 391.02939 found 391.02942.

*2-(benzo[d]oxazol-2-ylthio)-N-(5-(2-chlorophenyl)-1,3,4-thiadiazol-2-yl)acetamide* (**5b**). Yellow solid; yield: 61%, m. p. = 225–226 °C. ^1^H-NMR (300 MHz, DSMO-*d*_6_) δ 13.20 (s, 1H), 8.12 (dd, *J* = 6.8, 2.3 Hz, 1H), 7.69–7.49 m, 4H), 7.58–7.50 (m, 2H), 7.33 (dd, *J* = 6.0, 3.2 Hz, 2H), 4.55 (s, 2H). ^13^C-NMR (75 MHz, DSMO-*d*_6_) δ 166.24, 163.38, 160.00, 158.04, 151.41, 141.11, 131.87, 131.08, 130.84, 130.58, 127.88, 124.72, 124.46, 118.35, 110.29, 35.40. HRMS (ESI) *m*/*z* calcd. for C_17_H_11_ClN_4_O_2_S_2_Na^+^ [M + Na]^+^ 424.99042 found 424.99051.

*2-(benzo[d]oxazol-2-ylthio)-N-(5-(3-chlorophenyl)-1,3,4-thiadiazol-2-yl)acetamide* (**5c**). Yellow solid; yield: 61%, m. p. = 248–249 °C. ^1^H-NMR (300 MHz, DSMO-*d*_6_) δ 13.23 (s, 1H), 7.98 (t, *J* = 1.5 Hz, 1H), 7.88 (dt, *J* = 6.9, 1.7 Hz, 1H), 7.64 (m, 2H), 7.57 (m, 2H), 7.35–7.30 (m, 2H), 4.55 (s, 2H). ^13^C-NMR (75 MHz, DSMO-*d*_6_) δ 166.21, 163.36, 160.72, 158.90, 151.41, 141.12, 134.02, 131.97, 131.29, 131.27, 126.23, 124.73, 124.47, 118.36, 110.31, 35.42. HRMS (ESI) *m*/*z* calcd. for C_17_H_11_ClN_4_O_2_S_2_Na^+^ [M + Na]^+^ 424.99042 found 424.99039.

*2-(benzo[d]oxazol-2-ylthio)-N-(5-(4-chlorophenyl)-1,3,4-thiadiazol-2-yl)acetamide* (**5d**). Yellow solid; yield: 60%, m. p. = 259–260 °C. ^1^H-NMR (300 MHz, DSMO-*d*_6_) δ 12.89 (s, 1H), 7.94 (d, *J* = 8.4 Hz, 2H), 7.60 (d, *J* = 2.3 Hz, 2H), 7.56 (d, *J* = 8.4 Hz, 2H), 7.33 (dd, *J* = 6.9, 1.9 Hz, 2H), 4.52 (s, 2H). ^13^C-NMR (75 MHz, DSMO-*d*_6_) δ 165.77, 162.67, 160.72, 158.21, 156.73, 151.08, 140.85, 134.88, 128.88, 128.17, 124.20, 123.97, 117.91, 109.71, 35.08. HRMS (ESI) *m*/*z* calcd. for C_17_H_11_ClN_4_O_2_S_2_Na^+^ [M + Na]^+^ 424.99042 found 424.99045.

*2-(benzo[d]oxazol-2-ylthio)-N-(5-(2-fluorophenyl)-1,3,4-thiadiazol-2-yl)acetamide* (**5e**). Yellow solid; yield: 60%, m. p. = 267–268 °C. ^1^H-NMR (300 MHz, DSMO-*d*_6_) δ 13.22 (s, 1H), 8.24 (t, *J* = 7.5 Hz, 1H), 7.68–7.56 (m, 3H), 7.49–7.38 (m, 2H), 7.33 (dd, *J* = 5.6, 3.5 Hz, 2H), 4.54 (s, 2H). ^13^C-NMR (75 MHz, DSMO-*d*_6_) δ 166.22, 163.33, 160.18, 159.90, 159.83, 156.88, 154.87, 154.77, 151.37, 141.07, 132.64, 132.53, 128.18, 125.37, 124.69, 124.42, 118.30, 116.60, 116.31, 110.25, 35.35. HRMS (ESI) *m*/*z* calcd. for C_17_H_11_FN_4_O_2_S_2_Na^+^ [M + Na]^+^ 409.01997 found 409.02002.

*2-(benzo[d]oxazol-2-ylthio)-N-(5-(3-fluorophenyl)-1,3,4-thiadiazol-2-yl)acetamide* (**5f**). Yellow solid; yield: 43%, m. p. = 243.4–244.5 °C. ^1^H-NMR (300 MHz, DSMO-*d*_6_) δ 13.22 (s, 1H), 7.78 (d, *J* = 8.4 Hz, 2H), 7.68–7.60 (m, 2H), 7.59–7.54 (m, 1H), 7.41–7.32 (m, 3H), 4.55 (s, 2H). ^13^C-NMR (75 MHz, DSMO-*d*_6_) δ 166.18, 163.97, 163.35, 160.73, 158.82, 151.40, 141.12, 132.22, 132.10, 131.48, 124.71, 124.44, 123.33, 118.34, 117.60, 117.33, 113.53, 113.22, 110.28, 35.46. HRMS (ESI) *m*/*z* calcd. for C_17_H_11_FN_4_O_2_S_2_Na^+^ [M + Na]^+^ 409.01997 found 409.02014.

*2-(benzo[d]oxazol-2-ylthio)-N-(5-(4-fluorophenyl)-1,3,4-thiadiazol-2-yl)acetamide* (**5g**). Yellow solid; yield: 62%, m. p. = 240.4–241.5 °C. ^1^H-NMR (300 MHz, DSMO-*d*_6_) δ 13.16 (s, 1H), 8.00 (dd, *J* = 8.7, 5.4 Hz, 2H), 7.64 (m, 2H), 7.39 (d, *J* = 8.8 Hz, 2H), 7.33 (d, *J* = 8.8, 5.4 Hz, 2H), 4.54 (s, 2H). ^13^C-NMR (75 MHz, DSMO-*d*_6_) δ 166.00, 163.33, 162.25, 158.19, 151.36, 141.09, 138.74, 131.29, 129.92, 129.18, 127.34, 124.67, 124.40, 124.05, 118.30, 110.24, 35.44. HRMS (ESI) *m*/*z* calcd. for C_17_H_11_FN_4_O_2_S_2_Na^+^ [M + Na]^+^ 409.01997 found 409.02008.

*2-(benzo[d]oxazol-2-ylthio)-N-(5-(2-bromophenyl)-1,3,4-thiadiazol-2-yl)acetamide* (**5h**). Yellow solid; yield: 55%, m. p. = 217–218 °C. ^1^H-NMR (300 MHz, DSMO-*d*_6_) δ 13.23 (s, 1H), 7.95 (dd, *J* = 7.7, 1.7 Hz, 1H), 7.82 (d, *J* = 7.9 Hz, 1H), 7.67–7.60 (m, 2H), 7.55 (t, *J* = 7.2 Hz, 1H), 7.59–7.43 (m, 1H), 7.32 (dd, *J* = 5.5, 3.7 Hz, 2H), 4.54 (s, 2H). ^13^C-NMR (75 MHz, DSMO-*d*_6_) δ 165.77, 162.91, 159.32, 159.08, 150.95, 140.65, 133.36, 131.57, 131.20, 130.40, 127.79, 124.27, 124.01, 120.91, 117.88, 109.84, 34.91. HRMS (ESI) *m*/*z* calcd. for C_17_H_11_BrN_4_O_2_S_2_Na^+^ [M + Na]^+^ 468.93990 found 468.94028.

*2-(benzo[d]oxazol-2-ylthio)-N-(5-(4-bromophenyl)-1,3,4-thiadiazol-2-yl)acetamide* (**5i**). Yellow solid; yield: 59%, m. p. = 246.4–248.2 °C. ^1^H-NMR (300 MHz, DSMO-*d*_6_) δ 13.18 (s, 1H), 7.89 (d, *J* = 8.4 Hz, 2H), 7.72 (d, *J* = 8.4 Hz, 2H), 7.67–7.60 (m, 2H), 7.33 (dd, *J* = 5.8, 3.2 Hz, 2H), 4.53 (s, 2H). ^13^C-NMR (75 MHz, DSMO-*d*_6_) δ 166.15, 163.34, 161.16, 158.63, 151.41, 141.11, 132.32, 129.23, 128.80, 124.73, 124.47, 124.04, 118.33, 110.29, 35.40. HRMS (ESI) *m*/*z* calcd. for C_17_H_11_BrN_4_O_2_S_2_Na^+^ [M + Na]^+^ 468.93990 found 468.94022.

*2-(benzo[d]oxazol-2-ylthio)-N-(5-(2-(trifluoromethyl)phenyl)-1,3,4-thiadiazol-2-yl)acetamide* (**5j**). Yellow solid; yield: 56%, m. p. = 188–190 °C. ^1^H-NMR (300 MHz, DSMO-*d*_6_) δ 13.24 (s, 1H), 7.96 (dd, *J* = 7.5, 1.5 Hz, 1H), 7.84–7.75 (m, 3H), 7.68–7.61 (m, 2H), 7.36–7.31 (m, 2H), 4.55 (s, 2H). ^13^C-NMR (75 MHz, DSMO-*d*_6_) δ 166.35, 163.32, 159.80, 158.49, 151.39, 141.08, 132.74, 130.89, 128.15, 126.84, 125.35, 124.69, 124.42, 118.30, 110.25, 35.36. HRMS (ESI) *m*/*z* calcd. for C_18_H_11_F_3_N_4_O_2_S_2_Na^+^ [M + Na]^+^ 459.01677 found 459.01675.

*2-(benzo[d]oxazol-2-ylthio)-N-(5-(3-(trifluoromethyl)phenyl)-1,3,4-thiadiazol-2-yl)acetamide* (**5k**). Yellow solid; yield: 57%, m. p. = 252–253 °C. ^1^H-NMR (300 MHz, DSMO-*d*_6_) δ 13.26 (s, 1H), 8.23 (d, *J* = 11.8 Hz, 2H), 7.89 (d, *J* = 7.9 Hz, 1H), 7.77 (t, *J* = 7.7 Hz, 1H), 7.69–7.57 (m, 2H), 7.33 (dd, *J* = 6.0, 3.2 Hz, 2H), 4.55 (s, 2H). ^13^C-NMR (75 MHz, DSMO-*d*_6_) δ 166.22, 163.30, 160.65, 159.01, 151.36, 141.08, 131.06, 131.02, 130.62, 130.59, 130.22, 126.97, 124.66, 124.40, 122.98, 118.29, 110.22, 35.38. HRMS (ESI) *m*/*z* calcd. for C_18_H_11_F_3_N_4_O_2_S_2_Na^+^ [M + Na]^+^ 459.01677 found 459.01691.

*2-(benzo[d]oxazol-2-ylthio)-N-(5-(4-(trifluoromethyl)phenyl)-1,3,4-thiadiazol-2-yl)acetamide* (**5l**). Yellow solid; yield: 61%, m. p. = 269.5–271 °C. ^1^H-NMR (300 MHz, DSMO-*d*_6_) δ 13.27 (s, 1H), 8.15 (d, *J* = 7.9 Hz, 2H), 7.87 (d, *J* = 7.9 Hz, 2H), 7.69–7.57 (m, 2H), 7.32 (dd, *J* = 5.5, 3.4 Hz, 2H), 4.55 (s, 2H). ^13^C-NMR (75 MHz, DSMO-*d*_6_) δ 169.78, 165.90, 158.80, 151.14, 146.37, 140.88, 133.57, 127.27, 125.68, 124.21, 123.99, 117.94, 116.51, 109.73, 35.06. HRMS (ESI) *m*/*z* calcd. for C_18_H_11_F_3_N_4_O_2_S_2_Na^+^ [M + Na]^+^ 459.01677 found 459.01685.

*2-(benzo[d]oxazol-2-ylthio)-N-(5-o-tolyl-1,3,4-thiadiazol-2-yl)acetamide* (**5m**). Yellow solid; yield: 60%, m. p. = 217.6–218.7 °C. ^1^H-NMR (300 MHz, DSMO-*d*_6_) δ 13.11 (s, 1H), 7.64 (dd, *J* = 11.7, 6.9 Hz, 3H), 7.41 (d, *J* = 6.0 Hz, 2H), 7.38–7.32 (m, 3H), 4.54 (s, 2H), 2.49 (s, 3H). ^13^C-NMR (75 MHz, DSMO-*d*_6_) δ 166.06, 163.35, 161.61, 158.68, 151.37, 141.08, 136.34, 131.44, 130.05, 130.01, 129.09, 126.40, 124.68, 124.41, 118.31, 110.24, 35.35, 21.07. HRMS (ESI) *m*/*z* calcd. for C_18_H_14_N_4_O_2_S_2_Na^+^ [M + Na]^+^ 405.04504 found 405.04507.

*2-(benzo[d]oxazol-2-ylthio)-N-(5-m-tolyl-1,3,4-thiadiazol-2-yl)acetamide* (**5n**). Yellow solid; yield: 59%, m. p. = 240–241 °C. ^1^H-NMR (300 MHz, DSMO-*d*_6_) δ 13.12 (s, 1H), 7.78–7.69 (m, 2H), 7.68–7.59 (m, 2H), 7.40 (t, *J* = 7.5 Hz, 1H), 7.36–7.24 (m, 3H), 4.53 (s, 2H), 2.38 (s, 3H). ^13^C-NMR (75 MHz, DSMO-*d*_6_) δ 166.00, 163.33, 162.25, 158.22, 151.36, 141.09, 138.74, 131.29, 129.92, 129.18, 127.34, 124.67, 124.40, 124.05, 118.30, 110.24, 35.44, 20.81. HRMS (ESI) *m*/*z* calcd. for C_18_H_14_N_4_O_2_S_2_Na^+^ [M + Na]^+^ 405.04504 found 405.04514.

*2-(benzo[d]oxazol-2-ylthio)-N-(5-p-tolyl-1,3,4-thiadiazol-2-yl)acetamide* (**5o**). Yellow solid; yield: 56%, m. p. = 248–249 °C. ^1^H-NMR (300 MHz, DSMO-*d*_6_) δ 13.10, 7.82 (d, *J* = 7.9 Hz, 2H), 7.65 (m, 2H), 7.33 (d, *J* = 8.5 Hz, 4H), 4.52 (s, 2H), 2.36 (s, 3H). ^13^C-NMR (75 MHz, DSMO-*d*_6_) δ 166.06, 163.37, 162.18, 158.14, 151.38, 141.10, 140.59, 129.88, 127.34, 126.85, 124.71, 124.44, 118.33, 110.27, 35.43, 20.97. HRMS (ESI) *m*/*z* calcd. for C_18_H_14_N_4_O_2_S_2_Na^+^ [M + Na]^+^ 405.04504 found 405.04514.

*2-(benzo[d]oxazol-2-ylthio)-N-(5-(3,4,5-trimethoxyphenyl)-1,3,4-thiadiazol-2-yl)acetamide* (**5p**). Yellow solid; yield: 57%, m. p. = 207.6–209 °C. ^1^H-NMR (300 MHz, DSMO-*d*_6_) δ 13.20 (s, 1H), 7.69–7.58 (m, 2H), 7.37–7.28 (m, 2H), 7.18 (s, 1H), 4.52 (s, 2H), 3.86 (s, 6H), 3.71 (s, 3H). ^13^C-NMR (75 MHz, DSMO-*d*_6_) δ 166.05, 162.10, 158.39, 153.37, 151.37, 141.12, 139.47, 139.18, 125.43, 124.72, 124.46, 118.34, 110.28, 104.25, 60.16, 56.08, 35.43. HRMS (ESI) *m*/*z* calcd. for C_20_H_18_N_4_O_5_S_2_Na^+^ [M + Na]^+^ 481.06108 found 481.06134.

*2-(benzo[d]oxazol-2-ylthio)-N-(5-(2-methoxyphenyl)-1,3,4-thiadiazol-2-yl)acetamide* (**5q**). Yellow solid; yield: 62%, m. p. = 250.2–251.4 °C. ^1^H-NMR (300 MHz, DSMO-*d*_6_) δ 12.93 (s, 1H), 8.27 (d, *J* = 7.8 Hz, 1H), 7.70–7.59 (m, 2H), 7.55–7.48 (m, 1H), 7.37–7.30 (m, 2H), 7.26 (d, *J* = 8.3 Hz, 1H), 7.13 (t, *J* = 7.4 Hz, 1H), 4.52 (s, 2H), 3.97 (s, 3H). ^13^C-NMR (75 MHz, DSMO-*d*_6_) δ 165.83, 163.42, 160.05, 156.60, 155.25, 151.36, 141.10, 131.83, 127.16, 124.69, 124.41, 121.14, 118.65, 118.29, 112.34, 110.24 (s), 56.01, 35.36. HRMS (ESI) *m*/*z* calcd. for C_18_H_14_N_4_O_3_S_2_Na^+^ [M + Na]^+^ 421.03995 found 421.03989.

*2-(benzo[d]oxazol-2-ylthio)-N-(5-(3-methoxyphenyl)-1,3,4-thiadiazol-2-yl)acetamide* (**5r**). Yellow solid; yield: 62%, m. p. = 258.2–259 °C. ^1^H-NMR (300 MHz, DSMO-*d*_6_) δ 13.16 (s, 1H), 7.70–7.60 (m, 2H), 7.47 (t, *J* = 9.2 Hz, 3H), 7.36–7.31 (m, 2H), 7.10 (d, *J* = 6.8 Hz, 1H), 4.54 (s, 2H), 3.84 (s, 3H). ^13^C-NMR (75 MHz, DSMO-*d*_6_) δ 166.11, 163.37, 162.04, 159.71, 158.46, 151.42, 141.14, 131.29, 130.56, 124.72, 124.46, 119.52, 118.35, 116.76, 111.45, 110.29, 55.34, 35.42. HRMS (ESI) *m*/*z* calcd. for C_18_H_14_N_4_O_3_S_2_Na^+^ [M + Na]^+^ 421.03995 found 421.04004.

*2-(benzo[d]oxazol-2-ylthio)-N-(5-(4-methoxyphenyl)-1,3,4-thiadiazol-2-yl)acetamide* (**5s**). Yellow solid; yield: 62%, m. p. = 262.7–264 °C. ^1^H-NMR (300 MHz, DSMO-*d*_6_) δ 13.06 (s, 1H), 7.87 (d, *J* = 8.7 Hz, 2H), 7.67–7.60 (m, 2H), 7.35–7.30 (m, 2H), 7.07 (d, *J* = 8.7 Hz, 2H), 4.52 (s, 2H), 3.82 (s, 3H). ^13^C-NMR (75 MHz, DSMO-*d*_6_) δ 165.93, 163.35, 161.94, 161.08, 157.68, 151.35, 141.08, 128.47, 124.68, 124.41, 122.51, 118.29, 114.70, 110.24, 55.36, 35.38. HRMS (ESI) *m*/*z* calcd. for C_18_H_14_N_4_O_3_S_2_Na^+^ [M + Na]^+^ 421.03995 found 421.03983.

*2-(benzo[d]oxazol-2-ylthio)-N-(5-(2,5-difluorophenyl)-1,3,4-thiadiazol-2-yl)acetamide* (**5t**). Yellow solid; yield: 57%, m. p. = 207–208 °C. ^1^H-NMR (300 MHz, DSMO-*d*_6_) δ 13.28 (s, 1H), 7.72–7.59 (m, 3H), 7.39–7.29 (m, 4H), 4.55 (s, 2H). ^13^C-NMR (75 MHz, DSMO-*d*_6_) δ 165.97, 162.93, 160.68, 160.60, 159.77, 157.33, 157.25, 150.98, 149.51, 140.67, 132.52, 124.30, 124.03, 117.91, 112.34, 112.05, 109.86, 34.94. HRMS (ESI) *m*/*z* calcd. for C_17_H_10_F_2_N_4_O_2_S_2_Na^+^ [M + Na]^+^ 427.01054 found 427.01041.

*2-(benzo[d]oxazol-2-ylthio)-N-(5-(2-chloro-5-fluorophenyl)-1,3,4-thiadiazol-2-yl)acetamide* (**5u**). Yellow solid; yield: 64%, m. p. = 207.7–209 °C. ^1^H-NMR (300 MHz, DSMO-*d*_6_) δ 13.31 (s, 1H), 7.70–7.60 (m, 3H), 7.58–7.53 (m, 1H), 7.49–7.42 (m, 1H), 7.37–7.30 (m, 2H), 4.56 (s, 2H). ^13^C-NMR (75 MHz, DSMO-*d*_6_) δ 166.38, 163.33, 161.91, 160.53, 158.58, 152.31, 151.38, 141.06, 134.01, 133.26, 133.15, 126.10, 124.68, 124.42, 118.31, 115.30, 115.00, 110.24, 35.34. HRMS (ESI) *m*/*z* calcd. for C_17_H_10_ClFN_4_O_2_S_2_Na^+^ [M + Na]^+^ 442.98099 found 442.98108.

*2-(benzo[d]oxazol-2-ylthio)-N-(5-(3,5-dimethylphenyl)-1,3,4-thiadiazol-2-yl)acetamide* (**5v**). Yellow solid; yield: 61%, m. p. = 261–262 °C. ^1^H-NMR (300 MHz, DSMO-*d*_6_) δ 13.12 (s, 1H), 7.69–7.60 (m, 2H), 7.55 (s, 2H), 7.33 (dd, *J* = 5.9, 3.2 Hz, 2H), 7.15 (s, 1H), 4.53 (s, 2H), 2.33 (s, 6H). ^13^C-NMR (75 MHz, DSMO-*d*_6_) δ 165.97, 163.32, 162.37, 158.10, 151.36, 141.08, 138.54, 132.02, 129.83, 124.66, 124.55, 124.39, 118.28, 110.22, 35.39, 20.71, 20.65. HRMS (ESI) *m*/*z* calcd. for C_19_H_16_N_4_O_2_S_2_Na^+^ [M + Na]^+^ 419.06069 found 419.06070.

### 3.5. Preparation of Aβ_25-35_

Aβ_25-35_ was purchased from Aladdin, Shanghai, China. Briefly, 1 mg of Aβ_25-35_ was dissolved in 1 mL of purified water. It was polymerized in a 37 °C incubator for 96 h to obtain an oligomeric Aβ_25-35_ fragment, which was frozen at −80 °C until use [14].

### 3.6. Cell Culture and Treatment

PC12 rat pheochromocytoma cells were obtained from the Laboratory of College of Life Science and Technology, China Pharmaceutical University. PC12 cells were maintained in RPMI 1640 (Gibco Life Technologies, Grand Island, NY, USA) supplemented with 10% FBS and 100 U/mL penicillin and grown in a humidified atmosphere of 5% CO_2_ at 37 °C. The culture medium was replaced every other day and passaged every two days. Cells were grown to about 80% confluence before treatment. Control cells were cultured under above conditions 24 h. The compound groups were treated with 20 μM Aβ_25-35_ and different concentrations of compounds, which were cultured together for 24 h. The donepezil groups were treated with 20 μM Aβ_25-35_ and 2.5 μg/mL donepezil together for 24 h.

### 3.7. Cell Viability

Cell viability was evaluated by MTT assay. Briefly, PC12 cells (1 × 10^5^ cells/well) were seeded into 96-well plates and grown to confluence. Then, they were treated with compounds in different concentrations for specified times. After drug treatment, 20 μL of MTT solution (5mg/mL) was added to each well for 3 h at 37 °C. After removal of the supernatant, 200 μL DMSO was added into each well and shaken at a low speed for 5 min on a shaker. The absorbance (OD) value was measured at 570 nm and 630 nm with a microplate reader (Thermo, Waltham, MA, USA). Cell viability was calculated from the OD value, which was expressed as the percentage of drug and control groups.

### 3.8. Western Blotting Analysis

Cells were cultivated onto 1.0 × 10^5^ cell/mL plates to grow. After treatment with compound **5c** and Aβ_25-35_ according to Section 3.6, cells were washed twice with PBS, and then lysed with RIPA buffer (containing 1% phosphatase inhibitors and 0.5% protease inhibitors) at 4 °C for 30 min. The supernatant was transferred to a 0.5 mL centrifuge tube and centrifuged at 12,000 rpm for 15 min at 4 °C. The protein content was determined by BCA and mixed with loading buffer, and finally denatured by heating in a boiling water bath for 10 min. The proteins were separated by 10% SDS-polyacrylamide gels, and then transferred onto a polyvinylidene fluoride (PVDF, Millpore, Temecula, CA, USA) membrane. PVDF membranes were blocked with 5% non-fat milk for over 2 h at room temperature, and incubated with respective primary anti-bodies at 4 °C overnight (the rabbit antibodies against Thr181-p-tau, Thr205-p-tau, Ser396-p-tau, Total-tau, GAPDH, p-GSK-3β, GSK-3β, p-Akt, Akt, iNOS, p-NF-κB, NF-κB were purchased from Cell Signaling Technology, Beverly, MA, USA; the rabbit antibodies against Bax, Bcl-2 and BACE1 were purchased from Abcam, Cambridge, UK). PVDF was washed 3 times with TBST and incubated with a horseradish peroxidase conjugated secondary antibody (goat anti-rabbit IgG/HRP were purchased from Bioss Antibodies, Beijing, China) for 1 h at room temperature. After the PVDF membranes were washed 3 times with TBST, the antibody-reactive bands were visualized with a gel imaging system (Clinx Science Instruments, Shanghai, China) using enhanced chemiluminescence detection reagents. The band intensity analysis was calculated by Image J software (version 1.49).

### 3.9. Zebrafish

Zebrafish (Danio rerio, AB line) were maintained in a recirculating aquaculture system (Beijing Aisheng Technology Co Ltd., Beijing, China). The fish were kept in the zebrafish system, with a light-dark cycle (14 h:10 h) and maintained at 28.5 ℃ ± 1 ℃. The fish water contained 5 mM NaCl, 0.17 mM KCl, 0.4 mM CaCl_2_ and 0.16 mM MgSO_4_. Fish spawning and fertilized embryos were collected from natural crosses of adult fish and cultured in an aquarium [58]. The zebrafifish was selected for this study and all animal experiments were performed in accordance with the guidelines issued by the Animal Ethics Committee (Association for Assessment and Accreditation of Laboratory Animal Care International (AAALAC) Certifificate NO.001458).

### 3.10. Mortality

Compound **5c** and donepezil solutions were prepared in DMSO, and the concentration gradients were diluted with fish water. The concentration of DMSO in chemical exposure was 0.2%. The embryos at 4 hpf were distributed into 6-well plate at around 10 larvae/well in 5 mL drug solution. Embryos were exposed to compound **5c** and donepezil at doses of 50, 100, 250, 500 μM (contains 0.2% DMSO) for 120 h postexposure, 0.2% DMSO was used as control to match the highest concentration of DMSO used in the treatments. The fish water (containing drug) was replaced every 24 h. The dead embryos were removed in order not to contaminate the surviving embryos. Three parallel replicates were performed.

### 3.11. Measuring Area of Pericardial Cavity

The embryos at 4 hpf were distributed into 6-well plate at around 15 embryos/well. Zebrafish embryos were treated with 50 and 100 μM of compound **5c** and donepezil. Zebrafish embryos from each exposure concentration were imaged at 96 hpf under a stereomicroscope. Heart morphology was analyzed using LAS V 4.0 software (Leica) and the pericardial cavities of the zebrafish were measured using Image J software. At least 15 individual embryos were imaged for each concentration at each time point. Embryos were monitored daily and their development, hatching rate and heart rate were observed at 96 hpf under a stereomicroscope.

### 3.12. Measuring Tactile Sensitivity

The embryos at 4 hpf were distributed into 6-well plate at around 15 embryos/well. Zebrafish embryos were treated with 50 and 100 μM of compound **5c** and donepezil for 96 hpf. Then, a circle with a diameter of 1 cm was drawn on the bottom of the dish. The zebrafish were placed in the center of the circle. The zebrafish were gently stimulated with a micro-injection, and the reactions after stimulation were observed. Each fish was stimulated 20 times.

### 3.13. Statistical Analysis

The values shown are expressed as mean ± standard error of the mean and statistical analysis was performed using SPSS 20.0 software. After the homogeneity of variance test, one-way ANOVA was used for data analysis among groups, and the statistical significance standard was *p* < 0.05, *p* < 0.01 or *p* < 0.001.

## 4. Conclusions

Among these benzo[d]oxazole-based derivatives, compound **5c** showed a good safety profile in zebrafish and PC12 cells. Compound **5c** also showed potential to inhibit Aβ_25-35_ activity and counteract apoptosis. Taken together, these results suggest that compound **5c** could be considered as a compound for further development of AD therapeutics.

## Figures and Tables

**Figure 1 molecules-25-05391-f001:**
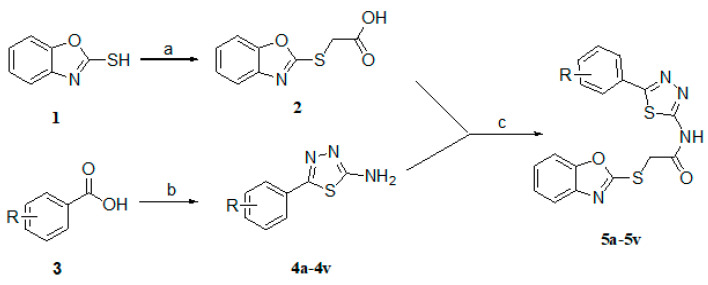
Preparation of benzo[d]oxazole-based derivatives.

**Figure 2 molecules-25-05391-f002:**
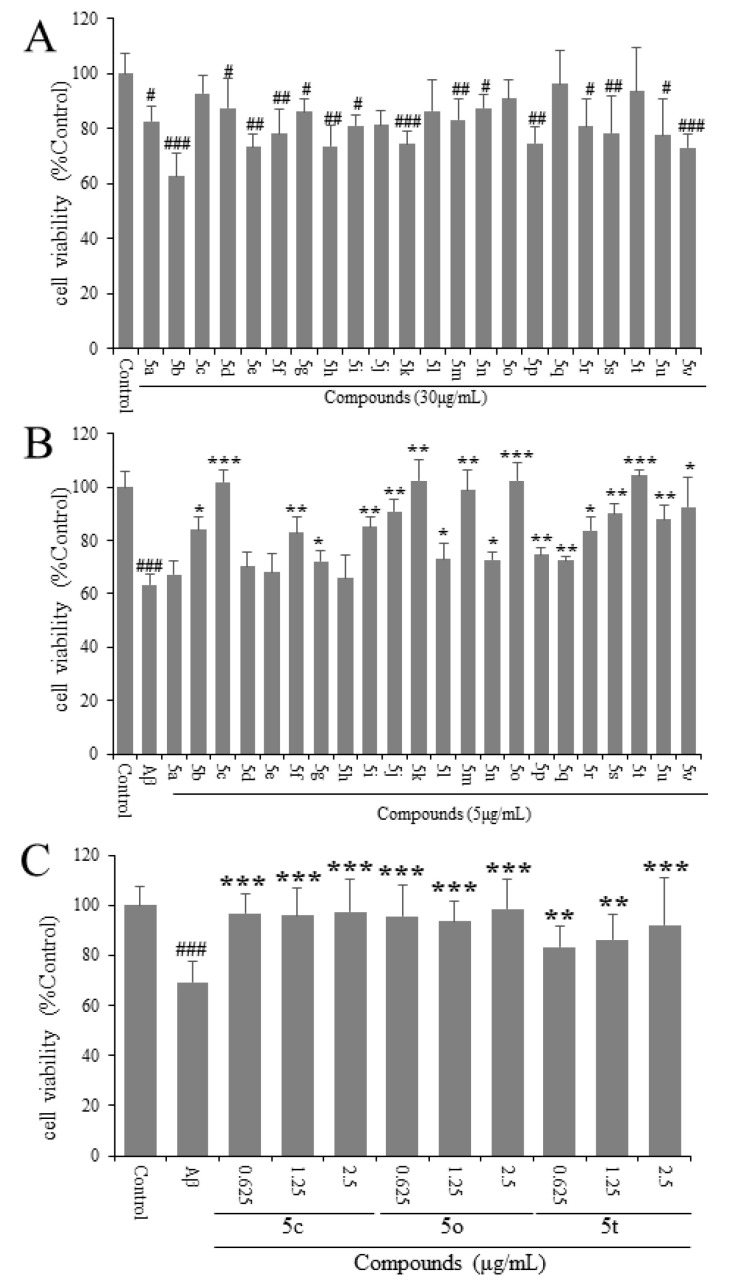
(**A**) Assessment of cytotoxicity of compounds **5a**–**5c** alone in PC12 cells. (**B**) PC12 cells were treated with compounds **5a**–**5c** and 20 μM β-amyloid (Aβ)_25-35_ for 24 h and cell viability was detected by MTT. (**C**) PC12 cells were treated with compounds **5a**–**5c** and 20 μM Aβ_25-35_ for 24 h and cell viability was detected by MTT. ^#^
*p* < 0.05, ^##^
*p* < 0.01, ^###^
*p* < 0.001 vs. control group; * *p* < 0.05; ** *p* < 0.01; *** *p* < 0.001 vs. model group.

**Figure 3 molecules-25-05391-f003:**
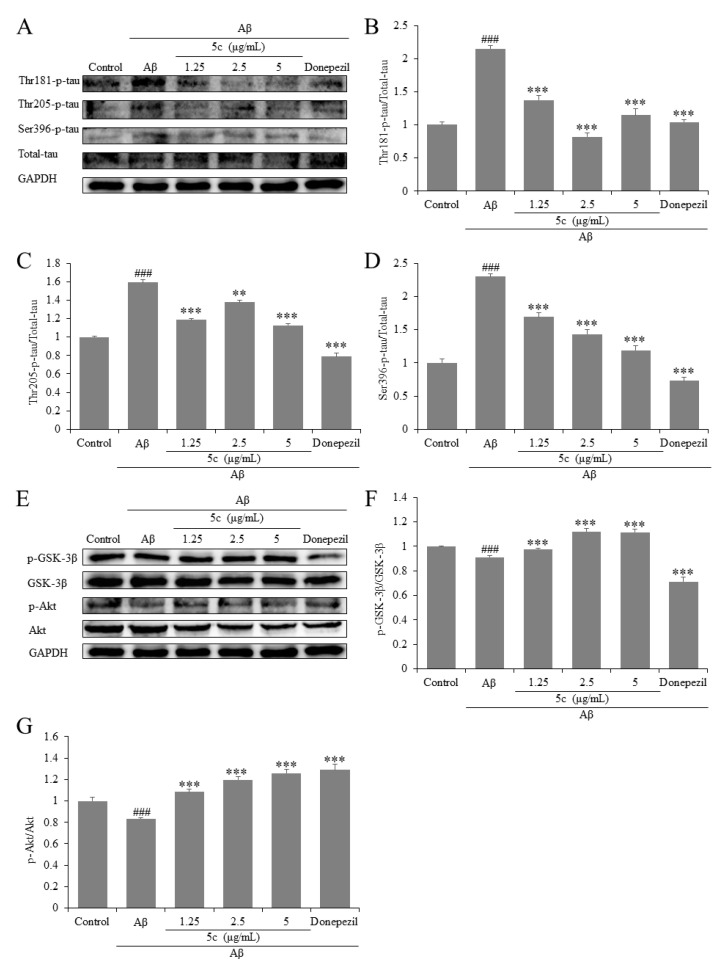
Phosphorylation effects of compound **5c** on tau, glycogen synthase kinase (GSK-3β) and Akt in Aβ_25-35_-induced PC12 cells. (**A**) The protein levels of tau were detected by Western blotting. Quantifications of Thr181-p-tau (**B**), Thr205-p-tau (**C**), and Ser396-p-tau (**D**) expression are presented in bar graphs, respectively. (**E**) The protein levels of GSK-3β and Akt were detected by Western blotting. Quantifications of GSK-3β (**F**) and Akt (**G**) expression are presented in bar graphs, respectively. ^###^
*p* < 0.001 vs. control group; ** *p* < 0.01; *** *p* < 0.001 vs. model group.

**Figure 4 molecules-25-05391-f004:**
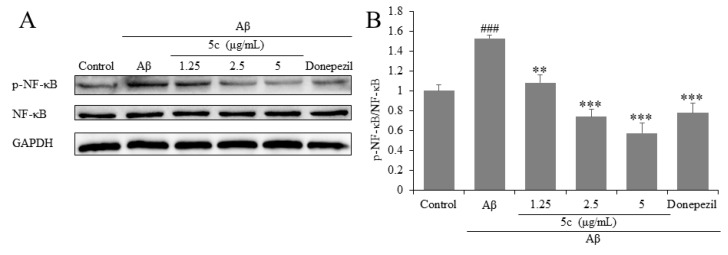
Effects on nuclear factor-κB (NF-κB) of compound **5c** in PC12 cells induced by Aβ_25-35_. (**A**) The protein levels of NF-κB were detected by Western blotting. (**B**) Quantification of NF-κB expression is presented in bar graphs. ^###^
*p* < 0.001 vs. control group; ** *p* < 0.01, *** *p* < 0.001 vs. model group.

**Figure 5 molecules-25-05391-f005:**
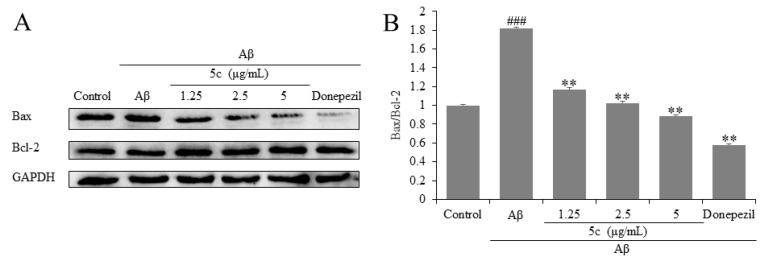
Effects on Bcl-2-associated X protein (Bax) and B-cell lymphoma 2 (Bcl-2) of compound **5c** in PC12 cells induced by Aβ_25-35_. (**A**) The protein levels of Bax and Bcl-2 were detected by Western blotting. (**B**) Quantifications of Bax/Bcl-2 expression are presented in bar graphs. ^###^
*p* < 0.001 vs. control group; ** *p* < 0.01 vs. model group.

**Figure 6 molecules-25-05391-f006:**
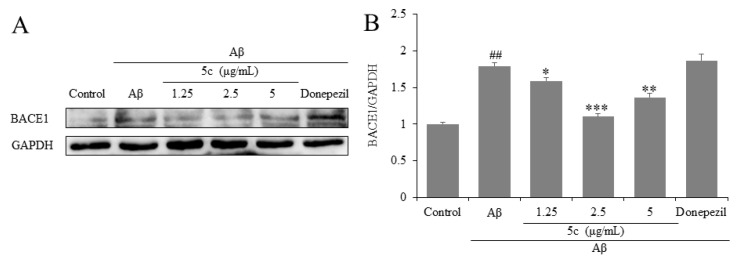
Effects on β-site amyloid precursor protein (APP)-cleaving enzyme 1 (BACE1) of compound **5c** in PC12 cells induced by Aβ_25-35_. (**A**) The protein levels of BACE1 were detected by Western blotting. (**B**) Quantification of BACE1 expression is presented in bar graphs. ^##^
*p* < 0.01 vs. control group; * *p* < 0.05, ** *p* < 0.01, *** *p* < 0.001 vs. model group.

**Figure 7 molecules-25-05391-f007:**
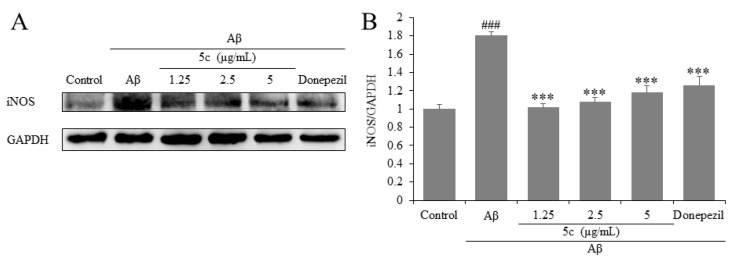
Effects on inducible nitric oxide synthase (iNOS) of compound **5c** in PC12 cells induced by Aβ_25-35_. (**A**) The protein levels of iNOS were detected by Western blotting. (**B**) Quantification of iNOS expression is presented in bar graphs. ^###^
*p* < 0.001 vs. control group; *** *p* < 0.001 vs. model group.

**Figure 8 molecules-25-05391-f008:**
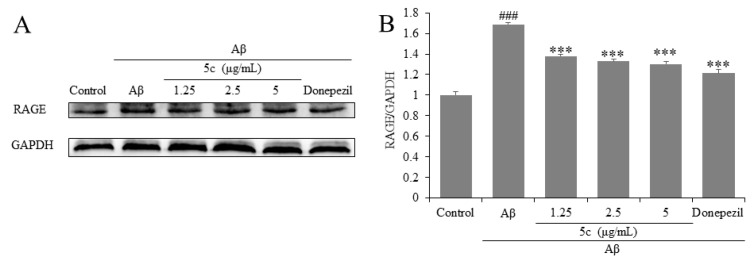
Effects on receptor for AGE (RAGE) of compound **5c** in PC12 cells induced by Aβ_25-35_. (**A**) The protein levels of RAGE were detected by Western blotting. (**B**) Quantification of RAGE expression is presented in bar graphs. ^###^
*p* < 0.001 vs. control group; *** *p* < 0.001 vs. model group.

**Figure 9 molecules-25-05391-f009:**
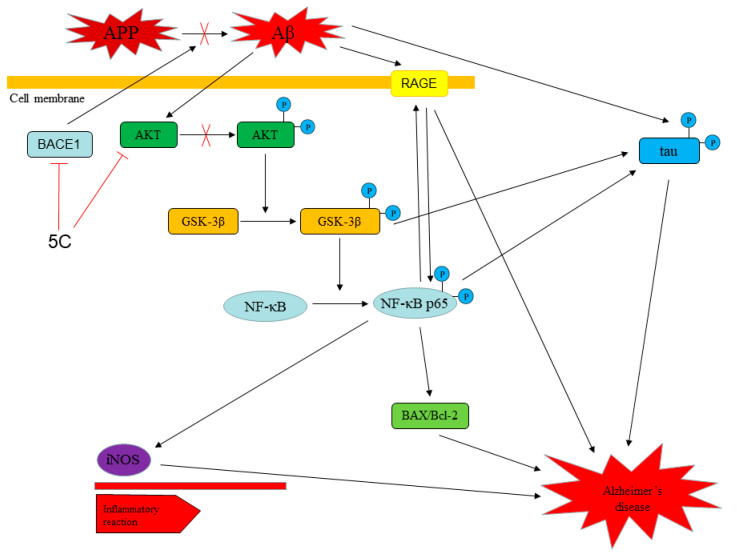
Schematic diagram of potential mechanisms by compound **5c** affects the activity of Aβ_25-35_-induced PC12 cells.

**Figure 10 molecules-25-05391-f010:**
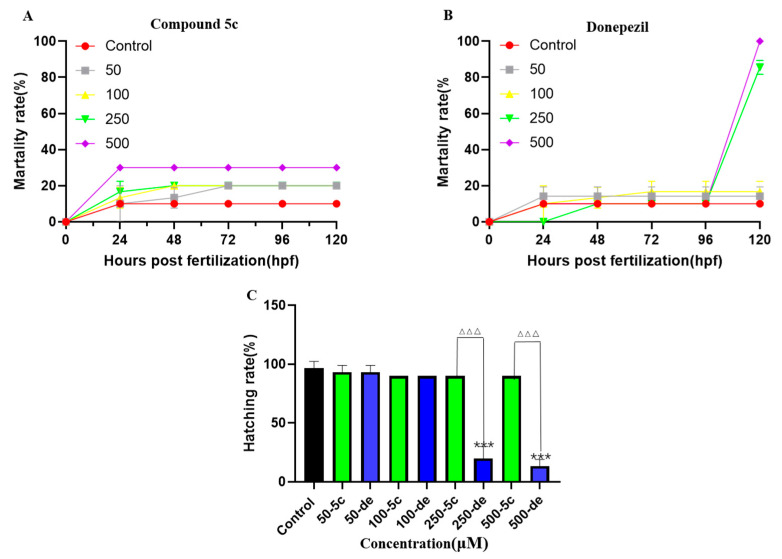
Effects of compound **5c** and donepezil on zebrafish mortality and hatching rate at 4 hours post-fertilization (hpf). Zebrafish embryos were exposed to different concentrations of compound **5c** and donepezil, and the mortality rate was recorded at 24, 48, 72, 96, and 120 hpf, respectively. The hatching rates were recorded for 120 hpf. (**A**) Effects of compound **5c** on zebrafish mortality. (**B**) Effects of donepezil on zebrafish mortality. (**C**) Effects of compound **5c** and donepezil on hatching rate. *** *p* < 0.001 vs. control group; ^△△△^
*p* < 0.001 compound **5c** group vs. donepezil group.

**Figure 11 molecules-25-05391-f011:**
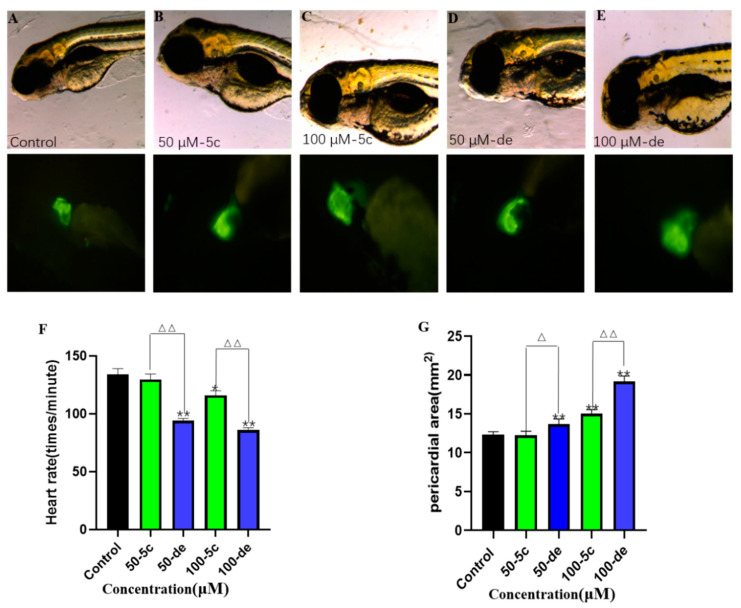
Effects of compound **5c** and donepezil on the heart rate and pericardial cavity area. The 4 hpf zebrafish embryos were exposed to compound **5c** and donepezil separately at 50 and 100 μM until 96 hpf. Representative image of effects of different compound **5c** and donepezil exposure groups on the pericardial cavity in zebrafish embryos: control (**A**), compound **5c** 50 μM (**B**), compound **5c** 100 μM (**C**), donepezil 50 μM (**D**), donepezil 50 μM (**E**), a histogram shows the difference between compound **5c** and donepezil in heart rate in zebrafish embryos (**F**), pericardial area of each treatment quantified (**G**). * *p* < 0.05, ** *p* < 0.01 vs. control group; ^△^
*p* < 0.05; ^△△^
*p* < 0.01 compound **5c** group vs. donepezil group.

**Figure 12 molecules-25-05391-f012:**
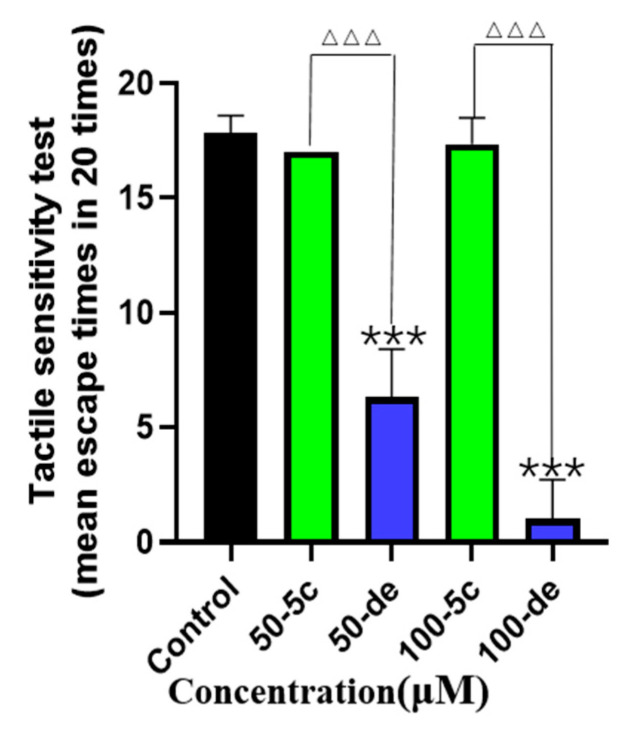
Effects of compound **5c** and donepezil on tactile sensitivity. The 4 hpf zebrafish embryos were exposed to compound **5c** and donepezil separately at 50 and 100 μM until 96 hpf, and tactile sensitivity was tested. *** *p* < 0.001 vs. control group; ^△△△^
*p* < 0.001 compound **5c** group vs. donepezil group.

**Table 1 molecules-25-05391-t001:** The synthetic routine of the title compounds **5a**–**5v**.

Compound	R	Compound	R	Compound	R
**5a**	H	**5i**	4-Br	**5q**	2-OCH_3_
**5b**	2-Cl	**5j**	2-CF_3_	**5r**	3-OCH_3_
**5c**	3-Cl	**5k**	3-CF_3_	**5s**	4-OCH_3_
**5d**	4-Cl	**5l**	4-CF_3_	**5t**	2,5-F_2_
**5e**	2-F	**5m**	2-CH_3_	**5u**	2-Cl-5-F
**5f**	3-F	**5n**	3-CH_3_	**5v**	3,5-(CH_3_)_2_
**5g**	4-F	**5o**	4-CH_3_		
**5h**	2-Br	**5p**	3,4,5-(OCH_3_)_3_

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
