# Peer review of "Design and Synthesis of New Benzo[d]oxazole-Based Derivatives and Their Neuroprotective Effects on β-Amyloid-Induced PC12 Cells"

_molecules, 2020, doi:10.3390/molecules25225391_

Round 1

Reviewer 1 Report

Specific points:

  1. Line 41. "breakdown of AChE by acetylcholine". Perhaps the opposite (i.e.
    breakdown of acetylcholine by AChE)?
  2. Fig. 3E,F. The authors should explain why Donezepil reduces p-GSK3beta levels in Abeta-treated PC12 cells in spite of its stimulatory effect on Akt activity (see G).
  3. Fig. 3F. Statistical significance is missing in the Donepazil column.
  4. A dedicated analysis of apoptosis rate in the various experimental conditions should be presented.
  5. Fig. 6B. Donezepil appears not to affect BACE1 levels in Abeta-treated PC12 cells? The authors should comment on.
  6. Line 151. "RAGE is a member of the immunoglobulin superfamily". Please quote Refs.
  7. Line 175. "compound 5c can decrease NF-κB levels". Perhaps, "compound 5c can decrease NF-κB activation levels"?
  8. Fig. 9. While NF-kappaB upregulates RAGE levels, RAGE signaling stimulates NF-kappaB in many cell types. Thus, the authors should add an arrow from RAGE to NF-kappaB in this cartoon.

Reviewer 2 Report

This is an exciting article that shows the promising value of substituted benzo[d]oxazole-based derivatives against Alzheimer's diseases. The work is well-executed. Noteworthy, I have one major concern and some minor points that should be addressed before publication:

Major point

- Point 3.4 The yield of the purified compounds is not very high (approx. 60%). Please show the effect of the used chemicals for compound 5c on the viability assays to check whether the reported protective role is not due to some of the compounds used during the chemical synthesis of these compounds.

Minor points

- Figure 2. The y-axe units are %, but the reported values vary from 0 to 1. Please correct

- Please indicate the correlation between the used units and the used micromolar concentration to treat cells to compare with those of B-amyloid.

- The authors showed several compounds with protective characteristics against amyloid B induced neurotoxicity, but they did not indicate the reasons to select compound 5c vs. the other compounds 5o and 5t. Please give a reasonable answer.

- The authors indicated that Compound 5c significantly inhibited the hyperphosphorylation of tau protein at Thr181-p-tau, Thr205-p-tau, 100 Ser396-p-tau. Based on the results shown in figure 3, this is only clear to me for the case of Thr181-p-tau. Please show better images that the selected ones to show the protective role of 5C against phosphorylation of Thr205-p-tau and  100 Ser396-p-tau. The authors also indicated that compound 5c increased the expression of Akt and GSK-3β. This is not a correct affirmation, but maybe the authors wanted to indicate that the ratio phosphorylated/non-phosphorylated Akt and GSK-3β increased. Please correct

- Line 186 The authors wrote 120hpf. Please add the name of the abbreviation in the text.

- This manuscript is focused on Alzheimer's disease. Therefore, to keep the coherence, the authors should include some data reporting the protective role of compound 5c in neuronal tissue vs. donezepil.

- Line 432. Please correct the word “dose”

Reviewer 3 Report

Manuscript in presented form is not recommended for publication. It is advised to split manuscript into two parts: chemical: synthesis, and biological: properties of a new compound invstigated in vitro and in vivo.  

Biological part, particularly western blotting assays, are  well performed and results are interesting, but description of experiments and results  presentation is not clear and requires major corrections and supplementations according to following comments.

General comments.

Manuscript is written in a rather confusing way, without clear planning and precision in presenting data and explanations of used experimental tools and  models. Many abreviations are used without explanation.

Introduction:

No clear explanation of the source of   β-amyloid (Aβ) peptides and their production mechanism in the cells  is given. No reasons for chosen cell and animal models and estimated parameters are presented. Important data and reasons for undertaken experiments are in the Result and Discussion section (references).

  1. „ breakdown of AChE by acetylcholine” - is other way around
  2. „Therefore, we selected the compounds for activity analysis of AD”- which compounds were tested for AD treatment ?
  3. Results and Discussion

Apoptosis was not estimated

2.2. Descriptions of experiments and results are very confusing and lack of important details. Many repetitions..

 Fig 2B. How long the cells were treated with Aβ25–35 before incubation with tested compounds?  In the text compounds 5a-5c were used, at the Fig.2B. all compounds. Why 5 μg/mL concentration was chosen?     

  1. „therapeutic effect „ -rather protective effect
  2. „pharmacological mechanism”- what mechanism?

2.3: „Akt and GSK-3β Activation”: expression and activation.

Fig 3, 4, 5, 6, 7, 8.. What are experimental conditions? No comments on donepezil treated samples. „represented as ratios (in percentage) of the control group”.What are control groups?

  1. „GSK-3β was the primary tau kinase and its activity required for serine dephosphorylation” : it needs explanation ( the role of PP2A).
  2. The role of iNOS is not explained. This is not pro-inflammatory cytokine producer.
  3. „In summary, compound 5c could suppress inflammatory responses via regulating NF-κB pathway, inhibit apoptosis in Aβ25-35-induced PC12 cells, and modulate the Akt/GSK-3β/NF-κB signaling pathway to delay AD” -  delay AD was not proven, only  prevention of Aβ25-35-induced PC12cytotoxicity.

2.9. Effects of compound 5c and donepezil on zebrafish vital parameters. There is no references to studies by the others.    

  1. Experimental section:

3.5. Preparation of Aβ25-35 oligomers; what is original procedure (reference)?

3.6. Cell Culture and Treatment: no description of the cells source (organism) is given (rat adrenal gland pheochromocytoma).What is a doubling time (change of medium every two days)? . Cell cultures were used as confluent or before confluency?

  1. What does it mean „normal conditions”?

3.8. Western Blotting Analysis: „inoculation” expression is used in microbiology.

410.” The drug-treated supernatant” ?

 No source and characteristic of used primary and secondary antibody,  neither the kind of chemiluminescence assay and analytical reader are given.

3.9 Zebrafish: source of animal, „fish spawning and fertilized embryos” collection-  original procedure ( reference)? The same concerns of the measurements of mortality, area of pericardial cavity and tactile sensitivity. Also physiological meaning of estimated parameters should be given.  

  1. „The embryos at 4 hpf were distributed into 6-well plate at around 15 larvae/well” - larvae are present from 7 dpf

Manuscript in presented form is not recommended for publication. It is advised to split manuscript into two parts: chemical: synthesis, and biological: properties of a new compound invstigated in vitro and in vivo.  

Biological part, particularly western blotting assays, are  well performed and results are interesting, but description of experiments and results  presentation is not clear and requires major corrections and supplementations according to following comments.

General comments.

Manuscript is written in a rather confusing way, without clear planning and precision in presenting data and explanations of used experimental tools and  models. Many abreviations are used without explanation.

Introduction:

No clear explanation of the source of   β-amyloid (Aβ) peptides and their production mechanism in the cells  is given. No reasons for chosen cell and animal models and estimated parameters are presented. Important data and reasons for undertaken experiments are in the Result and Discussion section (references).

  1. „ breakdown of AChE by acetylcholine” - is other way around
  2. „Therefore, we selected the compounds for activity analysis of AD”- which compounds were tested for AD treatment ?
  3. Results and Discussion

Apoptosis was not estimated

2.2. Descriptions of experiments and results are very confusing and lack of important details. Many repetitions..

 Fig 2B. How long the cells were treated with Aβ25–35 before incubation with tested compounds?  In the text compounds 5a-5c were used, at the Fig.2B. all compounds. Why 5 μg/mL concentration was chosen?     

  1. „therapeutic effect „ -rather protective effect
  2. „pharmacological mechanism”- what mechanism?

2.3: „Akt and GSK-3β Activation”: expression and activation.

Fig 3, 4, 5, 6, 7, 8.. What are experimental conditions? No comments on donepezil treated samples. „represented as ratios (in percentage) of the control group”.What are control groups?

  1. „GSK-3β was the primary tau kinase and its activity required for serine dephosphorylation” : it needs explanation ( the role of PP2A).
  2. The role of iNOS is not explained. This is not pro-inflammatory cytokine producer.
  3. „In summary, compound 5c could suppress inflammatory responses via regulating NF-κB pathway, inhibit apoptosis in Aβ25-35-induced PC12 cells, and modulate the Akt/GSK-3β/NF-κB signaling pathway to delay AD” -  delay AD was not proven, only  prevention of Aβ25-35-induced PC12cytotoxicity.

2.9. Effects of compound 5c and donepezil on zebrafish vital parameters. There is no references to studies by the others.    

  1. Experimental section:

3.5. Preparation of Aβ25-35 oligomers; what is original procedure (reference)?

3.6. Cell Culture and Treatment: no description of the cells source (organism) is given (rat adrenal gland pheochromocytoma).What is a doubling time (change of medium every two days)? . Cell cultures were used as confluent or before confluency?

  1. What does it mean „normal conditions”?

3.8. Western Blotting Analysis: „inoculation” expression is used in microbiology.

410.” The drug-treated supernatant” ?

 No source and characteristic of used primary and secondary antibody,  neither the kind of chemiluminescence assay and analytical reader are given.

3.9 Zebrafish: source of animal, „fish spawning and fertilized embryos” collection-  original procedure ( reference)? The same concerns of the measurements of mortality, area of pericardial cavity and tactile sensitivity. Also physiological meaning of estimated parameters should be given.  

  1. „The embryos at 4 hpf were distributed into 6-well plate at around 15 larvae/well” - larvae are present from 7 dpf

Round 2

Reviewer 1 Report

The authors have satisfied my previous criticism.

Author Response

Thanks for your previous criticism, and we have invited experts to make language promotion in the manuscript.

Reviewer 2 Report

The authors improved the quality of the manuscript. Noteworthy, I still have concerns regarding the protection/toxicity of the compound with such a low purity.  I strongly suggest checking the effect of the individual chemicals used for synthesis on the cell viability assays to know if the reported protective role is due to compound 5c or could be associated with any of the compounds used during the chemical synthesis.

Reviewer 3 Report

Suggested corrections and supplementations are introduced in proper manner. However manuscript still requires some language improvement  before publication.

Author Response

(The authors gave the same response as above.)
